

# Earth's core variability from the magnetic and gravity field observations

Anita Thea Saraswati[1,2], Olivier de Viron[1], and Mioara Mandea[2]

[1]Littoral, Environnement et Sociétés, La Rochelle Université and CNRS (UMR7266), La Rochelle, France
[2]Centre National d'Etudes Spatiales, 2 Place Maurice Quentin, 75039, Paris, France

**Correspondence:** Anita Thea Saraswati (anita.saraswati@univ-lr.fr)

**Abstract.** The motions of the liquid within the Earth's outer core lead to magnetic field variations together with mass distribution changes. As the core is not accessible to direct observation, our knowledge of the Earth's liquid core dynamics only relies on indirect information sources. Mainly generated by the core dynamics, the surface geomagnetic field provides information about the variations of the fluid motion at the top of the core. The dynamic of the fluid core is also associated with mass distribution changes inside the core, and produces gravitational field time fluctuations. By applying several statistical Blind Source Separation methods to both the gravity and magnetic field time series, we investigate the common space-time variabilities. We report several robust interannual oscillations shared by the two observation sets. Among those, a common mode of around 7 years looks very significant. Whereas the nature of the driving mechanism of the coupled variability remains unclear, the spatial and temporal properties of the common signal are compatible with a core origin.

## 1 Introduction

The Earth's magnetic field has been decreasing in strength over the past centuries, reducing by 10% over the last 150 years (Olson and Amit, 2006). The geomagnetic field is primarily generated by convective processes within Earth's iron-rich liquid outer core, which act like a dynamo (the geodynamo). Throughout Earth's history, the geomagnetic field has varied in strength and configuration on time scales ranging from years to billions of years (Lesur et al., 2022). These variations are related to deep-Earth's processes, and by understanding the full spectrum of these variations, we can explore the mechanisms driving the geodynamo.

Understanding the core dynamics involves a better understanding not only of the geomagnetic field and its variations but also of other possible observables. Indeed, our knowledge of the Earth's liquid core dynamics only comes from indirect sources of information. The dynamics of the core fluid change the magnetic field, apply a heterogeneous pressure field on the Core-Mantle Boundary (CMB) topography, deforming the inner Earth, moves density heterogeneities (Dumberry, 2010) – which gravitationally interact with the solid inner core and the mantle, and exchange angular momentum with the solid Earth through the electromagnetic, mountain and gravitational torques. In addition, the rheology property distribution inside the core affects the propagation of the seismic waves (Dehant et al., 2022, and reference therein). Observing the consequences of those inter-





actions, i.e. changes in the magnetic field, the Earth's shape and gravity field, and the Earth's rotation, can help to collect and
interpret pieces of information about the structure and dynamics of the inner core.

Analyzing such information sources in terms of core dynamics is a challenging task, as the Earth is a complex dynamic system, which implies that all those observables are sensible to many other sources of fluctuations. In particular, the climate dynamics dominate gravity, deformation, and Earth rotation change at most places and frequencies (e.g. Tapley et al., 2004; Rekier et al., 2022). Separating and understanding the core contributions in global magnetic and gravity data is the main
purpose of this study.

Separating the contributions from different sources can only be achieved by three different methods:

–  When one contribution is known with sufficient precision, it can be subtracted from the total signal, allowing to better detect and characterize the other contributions;

–  When two or more data sets are sensible to the same phenomena with different transfer functions, the joint analysis of
those data sets can allow to separate the contributions from the different phenomena;

–  When different contributions have different time-space signatures, statistical Blind Source Separation (BSS) Methods can be used to separate them.

Our paper combines the last two methods by applying join BSS methods on gravity and magnetic field time variations in order to evidence common dynamics. We test three different BSS methods – Principal Component Analysis (PCA, Preisendorfer
and Mobley (1988)), Singular Value Decomposition (SVD,von Storch and Zwiers (1999)), and Multivariate Singular Spectrum Analysis (M-SSA, Ghil et al. (2002)) – to assess the presence and the robustness of the retrieved signatures. The joint analysis of magnetic and gravity field time variations from this study aims at a more efficient separation of core contribution in the GRACE gravity data.

Chasing for core signatures in surface observation requires to use of long-term and global data sets, as the core dynamic
signatures are expected to be interannual and large-to-global scale (Lesur et al., 2022, and the reference therein). This is the reason we use long-term combined in-situ and satellite data sets, for both the gravity and the magnetic field, in the present study.

For the gravity field, we build on the time variable gravity fields from the GRACE/GRACE-Follow On missions. These missions allow us to retrieve monthly global gravity field from 2002 to the present, with a space resolution of a few hundred
kilometres. In addition, we also make use of another temporal gravity field based on the satellite laser ranging (SLR)/GRACE hybrid approach, which allows us to extend our analysis from 1992.

The data and methods applied in this study are described in Section 2. The separated time and spatial properties of the magnetic and gravity fields, obtained from each different analysis, are elaborately described in Section 3. Finally, in Section 4, we discuss the characteristics of the retrieved common modes, with regard to the literature on core dynamics, and we conclude
with the main arguments that support the thesis that these variations are coming from the processes of the Earth's deep interior.



## 2 Data and Methods

### 2.1 Data

#### 2.1.1 Geomagnetic field models

There have been significant breakthroughs in our understanding of rapid changes in the geomagnetic field over the past two
decades, mainly by the use of newly satellite measurements. Ørsted satellite was launched in 1999, followed by the CHAMP
and the SAC-C satellites in 2000. With the launch of the Swarm constellation, the geomagnetic field models resulting from
the mission provide new insights into Earth's interior. Indeed, these satellite data along with measurements obtained in the
worldwide geomagnetic observatory network offer the possibility to derive various geomagnetic field models of increasing
complexity and accuracy.

One of the most regularly updated main geomagnetic field models is the CHAOS series (Olsen et al., 2006), which provides
a high-resolution model and covers the past two solar cycles. Other main field models are also available that are developed by
other groups, such as the GRIMM series (e.g. Lesur et al., 2015), Comprehensive Model/Inversion series (e.g. Sabaka et al.,
2018), COV-OBS series (e.g. Huder et al., 2020), and the most recent one KALMAG model (Baerenzung et al., 2020). These
models are the product of a community effort and are frequently compared through the International Geomagnetic Reference
Field (IGRF) framework (e.g. Alken et al., 2021). A detailed summary and limitations of those models and also the modelling
techniques of the geomagnetic field can be consulted in Finlay (2020).

In the following, we present results based on two geomagnetic field models. They are COV-OBS.x2 (Huder et al., 2020)
(1840-2020), and CHAOS-7.12 (Finlay et al., 2020) (1998-2021). These models are built from a combination of ground-based
and satellite observations. The first and respectively the second derivatives in the radial direction of the core magnetic field
are known as secular variation (SV) and secular acceleration (SA). The SA of both models can be estimated on locations
of so-called Geomagnetic Virtual Observatories (Mandea and Olsen, 2006). Here, we consider the 10-degree grid (703 grid
points) using spherical harmonics up to degree 8. While the SA of CHAOS-7.12 can be computed directly from the spherical
harmonic coefficients, the SA of COV-OBS.x2 is calculated differently since the model is based on the projection onto splines
in the time domain of order 4 with 2 years of spacing knots. Thus, for COV-OBS.x2, we calculate the SV, at a yearly resolution.
Then, the monthly SA series is obtained by differentiating yearly SV and spline interpolation of the yearly series into monthly
resolution (Nicolas Gillet, personal communication). The linear trend of the time series is then removed to produce anomalies
of the geomagnetic field.

#### 2.1.2 Gravity field models

The tracking of the GRACE and GRACE-FO space gravity satellite pairs allows estimating the global Earth time-variable
gravity fields starting in 2002, with a monthly resolution (Kornfeld et al., 2019). The GRACE mission data analysis has
been successful in following the fluctuation of the surface water distribution associated with different hydrological processes
(e.g. Hassan and Jin, 2016; Rodell et al., 2018; Khaki and Awange, 2019; Frappart, 2020). GRACE also has improved our



knowledge of ocean dynamics (Landerer et al., 2015; Chen et al., 2020) and allows us to monitor the global change (Jeon et al., 2018; Tapley et al., 2019). Whereas the signal is strongly dominated by signatures associated with the climate system
dynamics – more than 90% of the signal comes from the climate system, only strong or coherent Earth interior signatures have been evidenced and analyzed, such as the Glacial Isostatic Adjustement (Sun and Riva, 2020, and reference therein), strong earthquakes and seismic cycle (Panet et al., 2018, for example), or even core processes and dynamics (Mandea et al., 2012, 2015).

Several centers have computed Earth's time variable gravity models based on the GRACE data: the Center for Space
Research (CSR, USA), the Jet Propulsion Laboratory (JPL, USA), the GeoForschungsZentrum (GFZ, De), the Groupe de Recherche en Géodésie (GRGS, France), the Goddard Space Flight Center (GSFC, USA), the University of Technology Gratz (TU Gratz, Au) (Flechtner et al., 2021; Landerer and Swenson, 2012; Tapley et al., 2005; Dahle et al., 2019; Kvas et al., 2019). A combined solution, COST-G, has also been developed (Peter et al., 2022). Most GRACE solutions are estimated in terms of spherical harmonic coefficients of the gravity potential every month - or every ten days -, whereas a few so-called mass
concentration (mascon) solutions, from CSR, JPL, GSFC, and GFZ, solve for the mass integrated over a set localized area of a few hundred square kilometers (Save et al., 2016; Watkins et al., 2015; Loomis et al., 2019). Higher-level products have also been developed and proposed, such as gridded equivalent water height, ocean bottom pressure, enhanced seasonal and trend, leakage-free separated ocean and continental gridded data, to only name a few.

This paper uses the IGG-SLR gravity field model (Löcher and Kusche, 2021), computed from GRACE leading empirical
orthogonal functions (EOFs) as the base functions when recovering the temporal gravity field from SLR. This SLR/GRACE hybrid approach provides us with Earth's gravity field time series for a period ranging from November 1992 to December 2020, whereas GRACE only started in 2002. For comparison, we also use GRACE RL06 Mascon Solutions (Rodell et al., 2004; Save, 2020). The time series are truncated within a period from September 2002 until August 2016 (168 months) to avoid the long gap between GRACE and GRACE-FO. For all the gravity field solutions, the spherical harmonic development
is limited to degree $nmax = 8$ and computed on the same grid points as the magnetic field, with a monthly resolution.

### 2.1.3 Data preprocessing

Before applying BSS techniques to the data sets, both the magnetic and gravity fields are pre-treated in order to smooth any sub-annual dynamics and produce anomalies of the fields. The linear trend, fit by the least-squares method, is subtracted from each point time series. For the gravity field, the seasonal cycle is then removed by subtracting the average of each month
(Hartmann and Michelsen, 1989). To remove the high-frequency signals, the time series of both fields are smoothed using a 13-month of moving average.

The time series is then normalized to a zero mean and a unit standard deviation by dividing each data sets by its corresponding standard deviation. Furthermore, anomalies at each grid are multiplied by the square root of the cosine of its latitude to take into account the weighting of the geographical grid size. While the modes are computed with the normalized data, we generate
the map with the full amplitude by de-normalize them back.





## 2.2 Methods

The geophysical data sets used in this study are given as gridded - longitude × latitude - values for each time step. The data set $\boldsymbol{X}$ thus has a dimension of $N \times D$, where $N$ is the time series length and $D$ is the number of grid points. The methods used here decompose the time-space variability $\boldsymbol{X}$ into modes consisting of time series, written here below as Principal Components 125 (PC) $e_k(t)$, and spatial patterns, also called load $A_k(\boldsymbol{p})$:

$$\boldsymbol{X}(\boldsymbol{p},t) = \sum_{k=1}^{K} A_k(\boldsymbol{p})e_k(t). \tag{1}$$

Those modes are obtained by computing the eigenvalues and eigenvectors of a covariance matrix, and the methods differ by the way this covariance matrix is built. The modes are ordered in decreasing order of the variance captured by the mode. Classically with such methods, most of the variance of the signal is captured by only a few modes. This allows for dimension 130 reduction of the data sets, by keeping only the modes that capture a significant amount of variance.

The statistical significance of the obtained modes is assessed by comparing the eigenvalues with those obtained from surrogate data sets with the same properties as the original data sets (Overland and Preisendorfer, 1982). Following Delforge et al. (2022), the surrogates are randomly generated as auto-regressive processes of order $p$, where $p$ is determined independently for each time series to minimize the Bayesian Information Criterion (BIC) and the coefficient fit on the time series. In this study, 135 the significance level of our Monte Carlo hypothesis test is set at the 95% level.

### 2.2.1 Principal component analysis (PCA)

In the PCA (Preisendorfer and Mobley, 1988), the covariance matrix of the data set is estimated, and the eigenvalues and eigenvectors of this matrix are computed. For joint PCA (see also Kutzbach, 1967), the two data sets, magnetic ($\boldsymbol{B}$) and gravity field ($\boldsymbol{G}$), are normalized and concatenated spatially ($\boldsymbol{X} = [\boldsymbol{B}\,\boldsymbol{G}]$).

### 140 2.2.2 Multivariate Singular Spectrum Analysis (M-SSA)

Singular Spectrum Analysis (SSA), first introduced by Broomhead and King (1986), is based on the Karhunen-Loève decomposition of stochastic processes into data-adaptive orthogonal functions. This analysis reconstructs the underlying complex dynamics from the time-delayed embedding temporal data sets (Ghil et al., 2002). The covariance matrix used for SSA is the lag-covariance matrix of a single time series, allowing to decompose a single time series into a sum of pseudo-periodic modes. 145 Oscillatory behaviour in SSA is captured in oscillatory pairs, which are formed from PCs with adjacent eigenvalues and similar frequencies that are in approximate phase quadrature (Plaut and Vautard, 1994; Ghil et al., 2002).

Applied to more than one time series, the so-called M-SSA uses a matrix composed of lag-covariance matrix of the different series. The details of the algorithmic can be found in Groth et al. (2017). The dimension of the data set is first reduced using PCA into $L$ channels (see Groth and Ghil, 2015). Each channel is embedded into an $M$-dimensional phase space to form $\mathbf{X}$-a 150 trajectory matrix of all channels, from which we obtain the matrix of size $LM \times N'$ where $N' = L - M + 1$. The M-SSA





method follows with calculating the singular value decomposition of $\mathbf{X}$ to obtain the space-time empirical orthogonal function (ST-EOF) and corresponding space-time principal component (ST-PC). The part of the original time series corresponding to a particular eigenmode is called the reconstruction component (RC), constructed from the corresponding ST-EOF and ST-PC. In M-SSA, a mode of oscillation is formed from a pair of eigenmodes. In this study, we also apply the varimax rotation of the

155 ST-EOFs to improve the separability of the patterns and frequencies (Groth and Ghil, 2011).

We then apply Monte Carlo hypothesis test against AR(1) noise to assess the statistical significance of the eigenvalues and the robustness of the obtained oscillatory pairs (Allen and Smith, 1996; Allen and Robertson, 1996). Following Groth and Ghil (2015), the proscrutes rotation of data time EOF (T-EOFs) is applied in the statistical analysis to avoid the risk of a too lenient significant test.

**2.2.3 Joint Singular value decomposition (SVD)**

The joint SVD technique works on decomposing the cross-covariance matrix of two different data sets that vary in space and time. This enables us to identify pairs of spatial patterns that capture the largest part of the common variability in the temporal domain. Cross-covariance matrix $C_{BG} = cov(\boldsymbol{B}, \boldsymbol{G}) = \boldsymbol{B}^T \boldsymbol{G}$ can be decomposed as $\text{SVD}(C_{BG}) = \mathbf{U}\mathbf{S}\mathbf{V}^T$. It generates two independent spatially uncorrelated sets of singular vectors, where $\boldsymbol{U}$ is the singular vectors of the left field, i.e. magnetic field

$\boldsymbol{B}$ and $\boldsymbol{V}$ is the singular vectors of the right field, i.e. gravity field $\boldsymbol{G}$, and a set of singular values $\boldsymbol{S}$ associated to the pairs of singular vectors. Detailed discussions of joint SVD analysis can be found in Bretherton et al. (1992), Wallace et al. (1992), and Venegas et al. (1997).

**2.2.4 Dominant period estimation**

For each mode, the dominant period (or frequency) is estimated as that of the maximum periodogram of that temporal prop-

170 erties. We apply the bootstrap technique to test the significance of the spectral power of the associated period (VanderPlas, 2018), in which the peak of the power spectrum is computed repeatedly on many random resamplings of the mode to estimate the distribution of that statistic (Ivezić et al., 2019).

Simulations are then performed to estimate the dominant period's uncertainty by adding normal-random phases to the time series in the Fourier domain to generate the surrogates with the same properties as the original time series (Schreiber and

175 Schmitz, 2000). We can then evaluate the distributions of the associated period. The period uncertainty is chosen as the standard deviation from the periods obtained in this simulation.



## 3 Results

### 3.1 Separated analysis of individual fields

Here, we focus on the results from COV-OBS.x2 and IGG-SLR. They cover longer observation/model periods, as required
by our analysis. The results obtained using the other data sets are shown in the appendix. To ease the reading, hereafter, the
COV-OBS.x2 model is called the magnetic field and IGG-SLR is mentioned as the gravity field.

As a first step, we analyze the magnetic and gravity fields in two separate individual computations using PCA and MSSA.
This allows us to analyze the space-time content of each data set without over-weighting the covariant part. Note that joint
SVD, by definition, cannot be used for separated analysis. We show the spatial pattern as a time correlation coefficient between
the PC (or the RC for MSSA) of that mode and the field variable at the same grid point as proposed by Wallace et al. (1992).
The significance of the Pearson correlation coefficients ($r$) is tested using the Student's $t$-test after evaluation of their number of
degrees of freedom from their auto-correlation function (Sciremammano, 1979; Von Storch and Zwiers, 2002). The locations
where the correlation is below 95% confidence level are marked with the white cross.

#### 3.1.1 Magnetic field

We have performed the PCA analysis on the normalized magnetic field model. From the applied Monte Carlo test (Vejmelka
et al., 2015), we found the first 14 modes from PCA to be significant (Fig. A1a), capturing together 99% of the total variance,
with the corresponding dominant periods between 3.5 - 24.4 years. The PC and spatial pattern of the first six-leading modes
obtained from PCA are shown in Fig. 1. The first mode (Fig. 1a), which captures 27% of the total variance, exhibits a time
variability with a period $T \approx 7.1$ years with an increase of amplitude in the recent years. This mode is significantly and strongly
correlated with the magnetic field around the equatorial band. Larger scale features are found around the Pacific Ocean and
Africa-Europe continent, while smaller features are exhibited around Central America and the Indian Ocean. This mode agrees
with the study by Gerick et al. (2021) and Gillet et al. (2022b), where they also identify a 7-year variation on the equatorial
band as the signature of Quasi-Geostrophic Magneto-Coriolis (QG-MC) eigenmodes in the fluid outer core, whereas Aubert
and Finlay (2019) and Aubert and Gillet (2021) attributed this variation to the Alfvén waves.

The second mode captures 20.6% of the total variance, with a period of $T \approx 24.4$ years, with the strongest correlations on
the Northern Pacific Ocean and on the Southern Ocean. The third mode (variance captured 14.6%) shows a decadal oscillation,
mostly active on the Atlantic and Southern Oceans, with a small active area close to Indonesia. The fourth PC has an oscillation
period of 6.8 year. The signal is mostly active in the western part of the Indian Ocean and around the South American continent.
PC 5, which accounts for 8.1%, has a dominant oscillation period of $T \approx 5.7$ years, similar to PC 6 which accounts for 4.8%
of variance. Both of those modes are separated with a lag of 1.58 years. Even though the dominant period is similar, the 2 PCs
have different spatial patterns and distinguishable eigenvalues according to the rule of thumb of North et al. (1982). PC 5 has
three lobes of stronger patterns on the southern low latitude and the Bay of Bengal, while the correlated patterns of the PC 6
are located around Central America and on the southern part of the Pacific and the Indian Ocean.




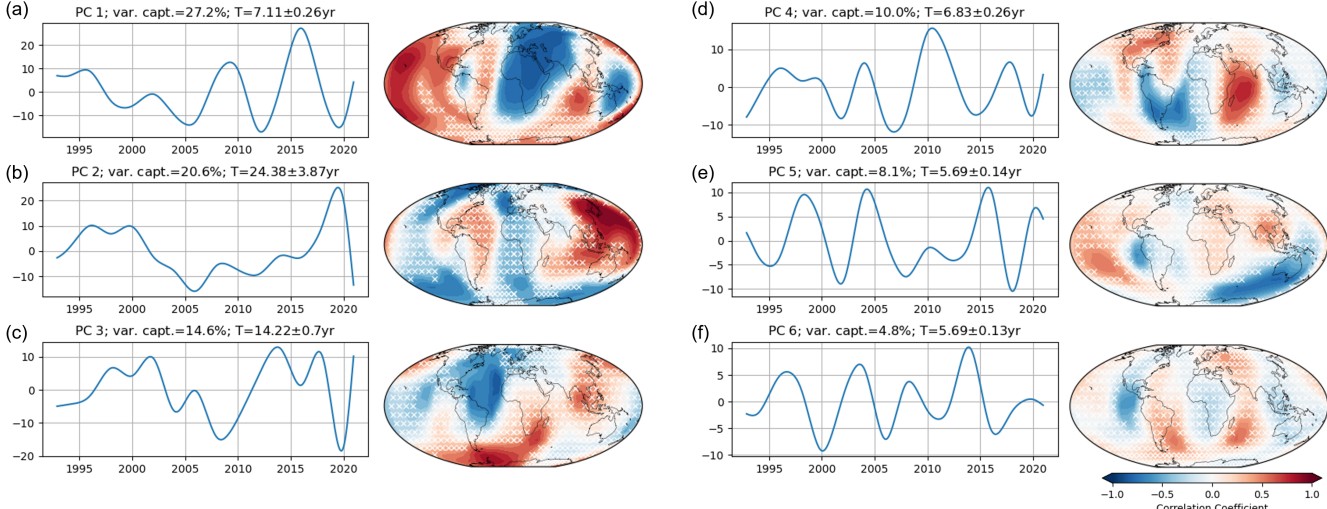

**Figure 1.** PCs of the magnetic field obtained from PCA (left). The corresponding correlation coefficients between the spatial patterns associated with each PC and the magnetic field are shown on the right. The white cross marks indicate the locations where the correlation significance does not reach the 95% level.

Unlike PCA, only three components are found significant at the 95% level in MSSA (Fig. A1b). The pair of ST-EOFs 1 and 2 represents an oscillation of $T \approx 7.1$ years, accounting for 37.29% of the total variance (Fig. 2a). The reconstruction of this mode shows an increase in variability as time advances. We found that the spatial patterns for this mode (Fig. 2c) are identical to the spatial patterns of the PC1 from PCA (Fig. 1a), with a spatial correlation of 0.97 between the two patterns.

ST-EOF 6 is also found significant at the 95% level. However, the pair of this component, ST-EOF 7, is only significant at the 90% level. Together, this pair constructs a mode with a period of 5.7 years that captures 15.05% of the total variance (Fig. 2c). The spatial pattern of this mode (Fig. 2d) resembles the spatial pattern of PC 5 in PCA (Fig. 1e), but with a stronger correlation on the Pacific area and no lobe on the Bengal.

In summary, two oscillatory modes appear to be robust in the magnetic field, with periods of $T \approx 7$ years and $T \approx 6$ years. Besides the temporal properties, the spatial patterns of these modes are also consistent in both techniques.

### 3.1.2 Gravity field

Similar procedures are applied to the analysis of the gravity field. The significance test in PCA leads us to keep 29 components which capture together up to 99% of the total variance (Fig. A5).

The first three modes (Fig. 3a-c) do not exhibit the oscillatory behaviour observed in the magnetic field modes. The first mode accounts for 40.5% of the total variance, forming a bidecadal variability, similar to a polynomial degree-2 of time. The second mode captures 18.7%. The third mode accounts for 15.4% of the gravity field variance. Areas with stronger and





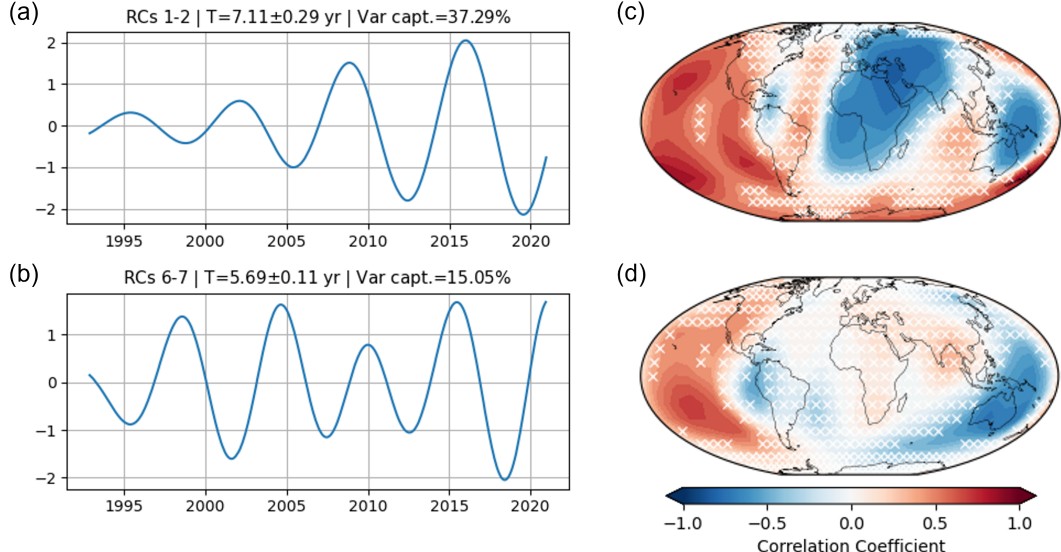

**Figure 2.** (a) Leading S-PC of RC1,2 that creates oscillation of 7.1 years, obtained from MSSA of the magnetic field. (b) Leading S-PC of RC6,7 form an oscillation of 5.7 years. As in Fig. 1, (c) and (d) show the correlation patterns of the mode 7.1 years and 5.7 years, respectively. The MSSA here uses a window length of $M = 110$ months. The white cross marks indicate the locations where the correlation significance does not reach the 95% level.

significant correlations are located mostly in the southern hemisphere, extending from the South of the Atlantic Ocean until the Western limit of the Pacific Ocean, and weaker correlations on the Asian continent.

    The fourth mode oscillation is dominated by a 7.1-year oscillation and captures 8% of the total variance. This mode strongly correlates around South and Central America, the Northern part of Africa close to the Gulf of Guinea, and extends from North to South along the meridian 100°E. The fifth mode (4.6%) is dominated by 8.5-year oscillations, with a smaller spatial

extent scattered across the oceans. The sixth mode has a variability of $T \approx 4.5$ years where the significant correlated areas are scattered all over the globe.

    From the Monte-Carlo test, we found 17 significant modes at the 95% level with MSSA (Fig. A5b). Among them, an oscillatory pair of ST-EOFs 5 and 6 construct a mode with $T \approx 6.8$ years (Fig. 4a). Compared to the spatial pattern of PC 4 from PCA (Fig. 3d) which has $T \approx 7.1$ years, the spatial pattern resulting from MSSA (Fig. 4b) is consistent with the spatial

pattern of PC 4 from PCA (Fig. 3d) which has $T \approx 7.1$ years, with a spatial correlation of 0.78.

    ST-EOFs pair 10 and 11 are also significant, showing a mode with a cycle of 3.88 years. The other significant ST-EOFs do not form oscillating pairs and correspond to higher frequencies. As they do not appear in the magnetic field, and considering their high frequency, we do not discuss them further in the present study.

    The time variability of the 7-year oscillatory modes of the gravity field resembles to some extent of those of the magnetic

field ($r = 0.78$), although significant differences are visible. The dynamic of higher frequencies is more clearly visible in the





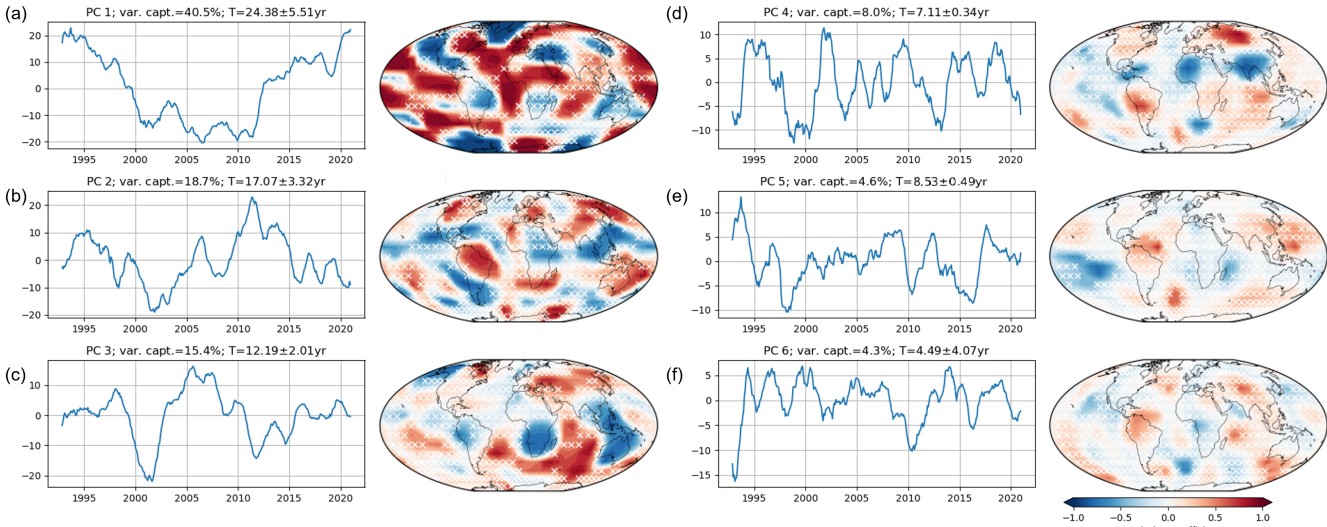

**Figure 3.** PCs of the gravity field obtained from PCA (left). The corresponding correlation coefficients between the spatial patterns associated with each PC and the gravity field are shown on the right. The white cross marks indicate the locations where the correlation significance does not reach the 95% level.

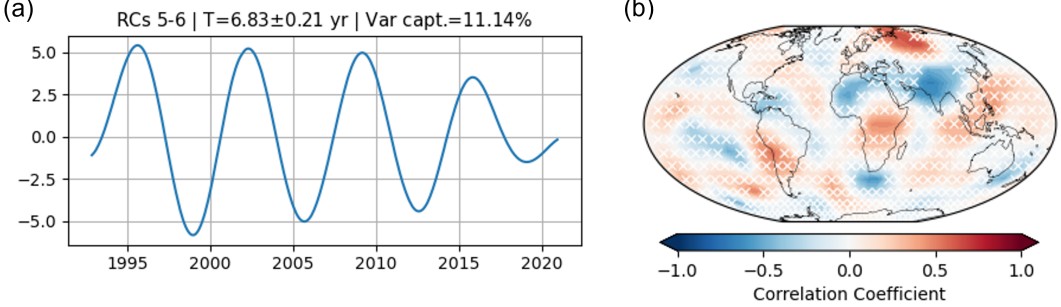

**Figure 4.** (a) Reconstructed component (RC) of the gravity field with ST-EOFs 5 and 6 obtained from MSSA method at period length of 6.8 year. (b) Correlation coefficient pattern between the gravity time series and the RCs in (a). The MSSA here uses a window length of $M =110$ months. The white cross marks indicate the locations where the correlation significance does not reach the 95% level.

gravity field, possibly coming from contamination from faster climate dynamics. While the oscillation amplitude increased with time on the magnetic field, those in the gravity field rather seem to decrease.

The spatial patterns differ between the magnetic and gravity field in the 7-year mode. We will return to the detail of the gravity field spatial pattern in the subsection 3.2.




## 3.2 Joint gravity-magnetic field analysis

We proceed with joint analyses of the magnetic and gravity fields to better highlight the similarities and differences between the two fields. For the joint PCA and MSSA, we concatenate the two normalized potential field data sets into a single multivariate time series. These methods generate common expansion coefficients (PC) of both fields and two spatial eigenvectors that are presented as correlation maps.

As in the previous section, we first test the significance of the eigenvectors (Fig. B1). The first 43 PCs of the joint PCA are significant against the normal random surrogates (Fig. B1a), which correspond to period lengths between 0.4 and 24.4 years. These significant components all together account for 99% of the total variance.

Figure 5 shows the results from the joint PCA. We find significant oscillatory modes that were identified in the PCA of the individual fields — bidecadal, decadal, ≈7, and ≈6 year. In the joint analysis, we always find PCs as a trade-off between that with the similar period from the analysis of the individual field Kutzbach (1967); Ghil et al. (2002). The associated spatial patterns are analogous to the spatial patterns from the PCA of the separate field with similar PC's period.

The dominant common variability between the two fields corresponds to a long-term behaviour, similar to polynomial degree-2 of time (PC1 in Fig. B2a) and to the PC1 of the gravity field (Fig. 3a). The spatial patterns are comparable to the ones resulting from the PCA of the individual field above — maps of PC2 for the magnetic field (Fig. 1b) and PC 1 for the gravity field (Fig. 3a).

The interannual variation of 6.8 years is captured by PC2, with 14.2% of the total variance captured. The PC of this mode resembles a trade-off of the separate PC in the individual fields of the associated period, i.e. dominant oscillations of 7 years found in the PC1 of the magnetic field with higher frequency dynamics from the gravity field. The spatial patterns of this mode are akin to the ones of PC1 in the magnetic field (Fig. 1a) and PC4 in the gravity field (Fig. 3d), except for the areas on South America and the Indian Ocean.

The third mode exhibits a time variability of 15.5 years. The resulting spatial patterns are consistent with the third mode from the separated analysis for the magnetic field (Fig. 1c) and the second mode for the gravity field (Fig. 3b). The fourth PC captures the third modes of the gravity field separated PCA (Fig. 3c) and of the magnetic field (Fig. 1c) with a cycle of 14.2 year, but both exhibit significantly different patterns with respect to that from the separated analysis.

From the joint MSSA, 7 ST-EOFs are identified as significant from the Monte Carlo test (Fig. B1b). ST-EOFs 3 and 5 are in phase quadrature and form an oscillatory pair of 7.4 year (Fig. 6a), which accounts for 21.8% of the variance captured. This mode period is consistent with that from the mode found in the above-mentioned MSSA analyses of the separated individual field. The mode amplitude increases with time, but is significantly less than in the separated MSSA of the magnetic field (Fig. 2a). The spatial patterns are similar to those from the separated fields.

An oscillatory pair with a period length of 6.1 year is also found to be significant, formed by the pair of ST-EOF 7 and 8 (Fig. 6b). This mode is consistent with the 6-year mode found in the magnetic field, where this period is not found in the MSSA of the gravity field.



**Figure 5.** (a-f) PCs obtained from PCA of the joint field. On the right part, the correlation map of the magnetic field and gravity field associated with each PC. The percentage of the variance captured by each PC is shown on the top of the time expansion. The portion of the variance captured in each field is mentioned at the top of the correlation map. The white cross marks indicate the locations where the correlation significance does not reach the 95% level.





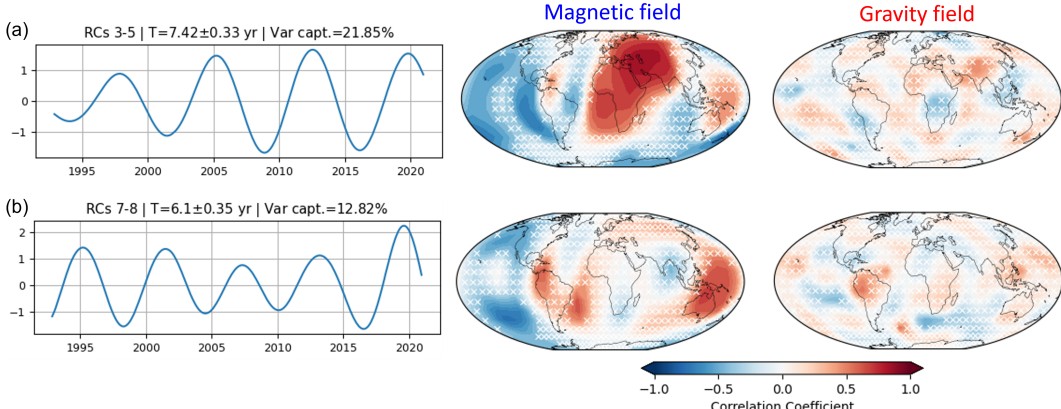

**Figure 6.** Reconstruction of the joint field of oscillatory pairs at a period length of 7.4 year (a) and 6.1 year (b). The correlation patterns of the magnetic and gravity field of each mode are given on the right side. The MSSA here uses a window length of $M =110$ months. The white cross indicates the areas with insignificant correlations at the 95% level.

Figure 7 shows the PCs associated with gravity and magnetic fields, separately, obtained from the joint SVD analysis. The first fourteen PCs are tested significant (Fig. B1c). Consistent with the PCA of the separate fields and the respective time variability of the gravity and magnetic fields, the temporal variations on the gravity field also contain a noticeably higher frequency than the magnetic field.

Bidecadal and decadal variabilities dominate the first three modes in the SVD analysis, with similar spatial patterns compared to the results of the PCA of the joint field. However, the time series length of 28 years that we use in this study limits the reliability of detecting such long-term variation, thus further elaboration on this behaviour is beyond the scope of this paper.

On the fourth mode, we find the oscillatory period of 7.1 years, with a temporal correlation coefficient $r = 0.58$. The spatial patterns of the magnetic field in this mode resemble the 7-year modes from other analyses. In contrast, the associated gravity field mode mixes that of periods 12.2 and 7 years from the separated analysis (Fig. 3c-d). The spatial patterns from this analysis are notably different from those from other analyses.

The fifth PC exhibits a dominant oscillation of $T \approx 6$ years, with captures 3% of the total variance. The spatial patterns found in this mode are comparable to the ones resulting from the joint MSSA (Fig. 6b), where the significant zones are consistent across these two different results.

The results in the joint field analyses are consistent with the ones in the separate analysis of the magnetic and gravity fields (subsection 3.1). A period of 7 years is detected in all analyses with consistent spatial patterns, in certain regions at least. The oscillation at a 6-year period is detected in all analyses except in the analysis of the gravity field. Despite the use of various types of analysis and input combinations, the space and time signatures of these modes exhibited sufficient similarities to support the validity of their detection. This suggests that the results are robust and reliable.





**Figure 7.** (a-f) The first six PCs of the magnetic (blue line) and gravity field (red line) obtained from the joint SVD technique. The corresponding dominant period and the Pearson's correlation coefficient ($r$) are written in the legend. The correlation patterns of the magnetic and gravity field of each mode are given on the right side. The white cross indicates the areas with insignificant correlations at the 95% level.

## 4 Discussion and Conclusions

We used the co-analysis of magnetic and gravity fields to separate between climate-induced and internal – probably core – signatures in the gravity field data. The application of different techniques also allows us to mine for common behaviour between magnetic and gravity fields and to assess the robustness of the associated principal components of the time series. The




consistency of those common behaviours over different data sets further demonstrates the robustness of those signatures and confirms the obtained time series and space patterns.

The applied analyses provide rich information about the temporal and spatial behaviour of the magnetic and gravity fields. In the following, we summarize these results in two dedicated figures.

A summary of significant mode periods is displayed in Fig. 8. As expected, more modes with short-term variability (of the order of a couple of years) are found in the gravity-only-based analysis. The dynamics of the mass redistribution on the Earth's surface are represented in the modes with higher frequencies (Gruber et al., 2011), which are mainly related to the climate system time variability. With a maximum series length of 28 years, we focus here on oscillations with periods longer than 4 years and shorter than 14 years.

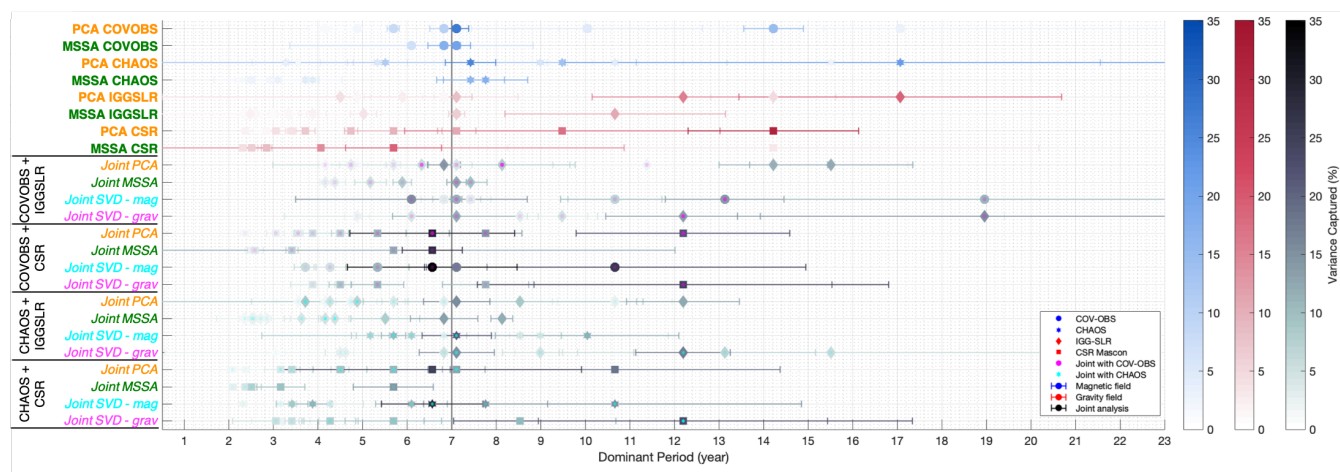

**Figure 8.** Significant mode periods from each analysis, along with the uncertainty of the estimated dominant period. The error bars show the corresponding period length error estimate ($1\sigma$). The methods and data sets are listed in the y-axis: PCA (orange), MSSA (green), and Joint SVD (magenta and cyan). Individual field analyses are indicated in bold text, and joint field analyses are in italics. Each data set is displayed by different symbols. The blue, red, and black represent the results of magnetic, gravity, and joint fields, respectively. The colour bars show the percentage of the variance captured by each mode.

Modes within a period range of 6.5-7.5 years are found significant in 20 analyses out of 24. In the following, this is named as the "7-year mode" and it captures in average 13.8% of the variance, with a maximum of 35.1% in the PCA of the COV-OBS.x2. The amplitude evolution of the PCs detected in the separated analysis and in the SVD exhibit differences: an increase for the magnetic field (Fig. 1, 2), and a slight decrease for the gravity field (Fig. 3, 4, 7).

Unlike M-SSA, SVD and PCA do not favour pseudo-periodic behaviour. Finding time oscillations in SVD and PCA results is thus a piece of evidence that this periodic behaviour is significant in both time series.

Besides the 7-year mode, oscillations with period $T \approx 6$ years are also found significant, appearing in 16 analyses out of 24. Taking into account the uncertainty of the period estimates and also the frequency resolution of the spectrum (Lathi and Green, 2005, e.g.), it is not possible to exclude that the modes at periods 6 and 7 years correspond to the same physical phenomena.





However, considering that the 6-year oscillation mostly occurs simultaneously with the 7-year one and that their spatial patterns
are different (Fig. C1), they are more probably the signatures of distinct phenomena.

The areas defined in Fig. 9 as significant for the magnetic field are close to those indicated in different studies and related to
a 7-year oscillation (e.g. Buffett and Matsui, 2019; Aubert and Gillet, 2021; Gillet et al., 2022a, b). Our results coincide with
those of previous studies and confirm our approach. Consequently, we elaborate no more on the magnetic aspect in the present
paper.

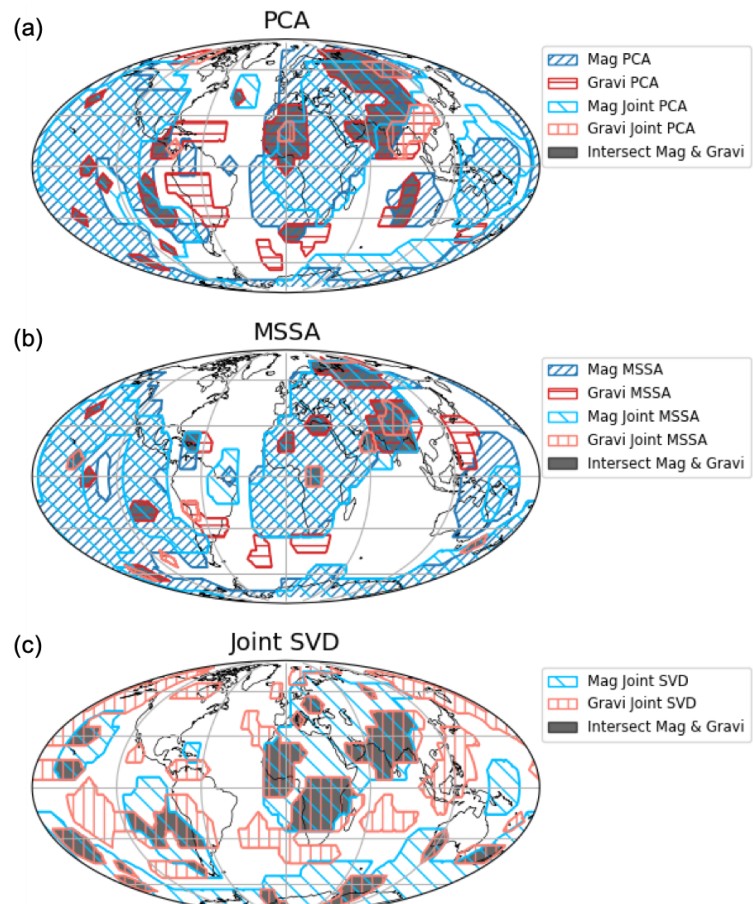

**Figure 9.** Maps of the areas associated with the 7-year mode where the correlation coefficients between the magnetic field (blue) or the
gravity field (red) and the obtained time PC from PCA (a), MSSA (b), and Joint SVD (c) are significant at the 95% level. The areas where
both the magnetic and gravity fields are significantly correlated with the 7-year mode are marked in grey.

Figure 9 shows locations for the 7-year mode of the gravity and magnetic fields, and underlines those where both are
significant, without however a clear correlation between those two space patterns. This is not surprising, considering that the
transformation of core processes into mass and into magnetic anomalies are different, and probably rely on different properties



of the core and of the CMB. In addition, the 7-year mode in the gravity field might still be influenced by residual signals from surface processes, leading to differences in spatial behaviour reflected in the gravity field.

Some possible mechanisms of the dynamic core processes that can perturb the gravity field have been previously proposed: changes in the density field within the volume of the core (Dumberry, 2010), dissolution-crystallization process at the CMB (Mandea et al., 2015), pressure anomalies at the CMB that entrain the elastic deformation in the Earth (see Greff-Lefftz et al., 2004; Dumberry and Bloxham, 2004; Dumberry, 2010), and the reorientation of the inner core along with its lateral heterogeneity (Gillet et al., 2021; Dumberry and Mandea, 2022). However, the quantification of the gravitational perturbation

due to those proposed mechanisms remains challenging, particularly in elucidating the perturbations at such a scale as that of the gravity field patterns. Further investigation is necessary, for example, estimating the gravitational effect of the core dynamics, particularly in the interannual time scale and on higher harmonic degrees. Building complete models of such motions is beyond the scope of this paper.

    The results presented here are encouraging to look for information on the dynamics of the Earth's core in other data sets, such as the gravity field, which might provide ancillary input as a base to build models that enhance our understanding of the

340 properties and dynamics of the core. Over the upcoming years, longer magnetic and gravity observations will be available, allowing for a better separation between climate and core-induced signatures. More refined works in separating the sources of the observed magnetic and gravity fields with taking into account the physical properties might also help to isolate better and understand different components in the Earth's core system.

*Code availability.* The geomagnetic time series are computed using ChaosMagPy (https://doi.org/10.5281/zenodo.3352398). The time series of the gravity anomalies are generated using a modified version of gravity-toolkit (Sutterley, 2023) by H. Lecomte. The autoregressive model fitting was analysed using statsmodel package (Seabold and Perktold, 2010).

*Author contributions.* AS, OdV, and MM contributed to the design and implementation of the research, the analysis of the results and the writing of the manuscript.

*Acknowledgements.* We would like to thank N. Gillet and D. Jault for fruitful discussions. The work is performed under the framework of the GRACEFUL project of the European Research Council (grant no. 855677). This study was supported by CNES as an application of the space gravity and magnetism missions.



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



# Appendix A: Analysis of individual fields

## A1   Magnetic fields

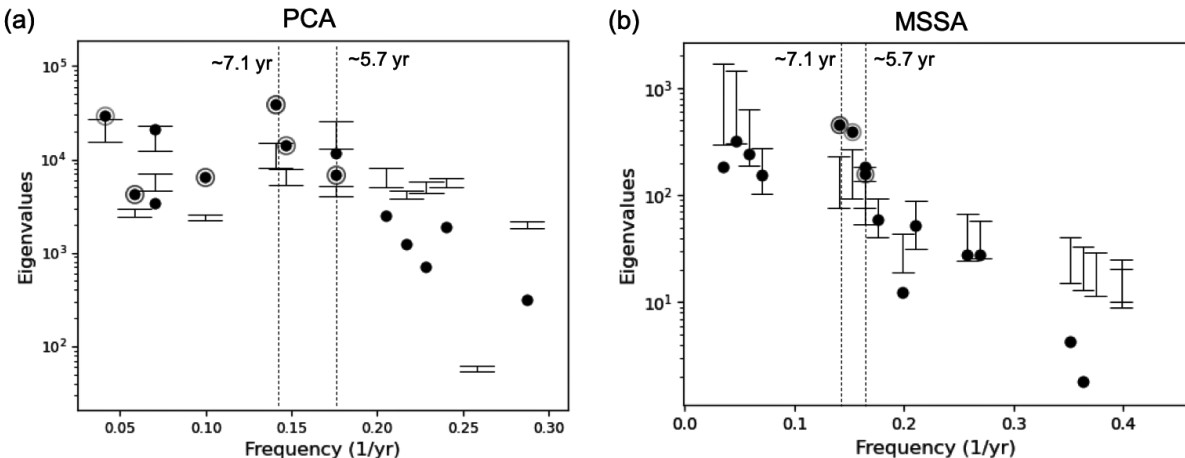

**Figure A1.** Significant test of PCs using Monte Carlo-type hypothesis. (a) Comparison of eigenvalues in PCA analysis between COV-OBS.x2 and surrogates based on AR(p). (b) Spectral properties of COV-OBS.x2 obtained from MSSA, with a subsequent varimax rotation use ST-EOFs 1-13. The estimated eigenvalues are plotted in black dots as a function of their corresponding frequency. The lower and upper ticks on the error bars indicate 5% and 95% of percentiles from a Monte Carlo test with scaled proscrutes target rotation of T-EOFs (Groth and Ghil, 2015). The significant PCs are circled.




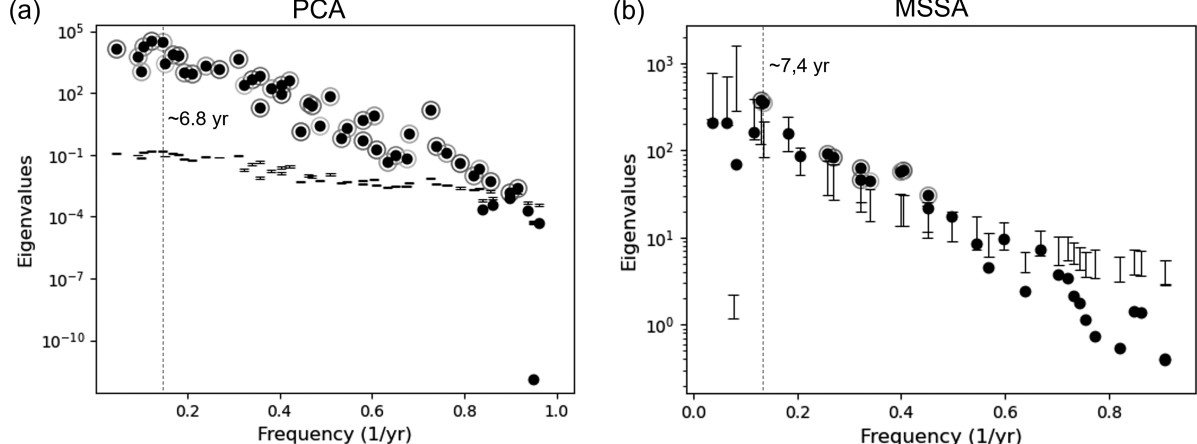

**Figure A2.** Significant test of PCs using Monte Carlo-type hypothesis. (a) Comparison of eigenvalues in PCA analysis between CHAOS-7.12 and surrogates based on AR(p). (b) Spectral properties of CHAOS-7.12 obtained from MSSA, with a subsequent varimax rotation use ST-EOFs 1-21. The estimated eigenvalues are plotted in black dots as a function as their corresponding frequency. The lower and upper ticks on the error bars indicate 5% and 95% of percentiles from a Monte Carlo test with scaled proscrutes target rotation of T-EOFs (Groth and Ghil, 2015). The significant PCs are indicated in circle.





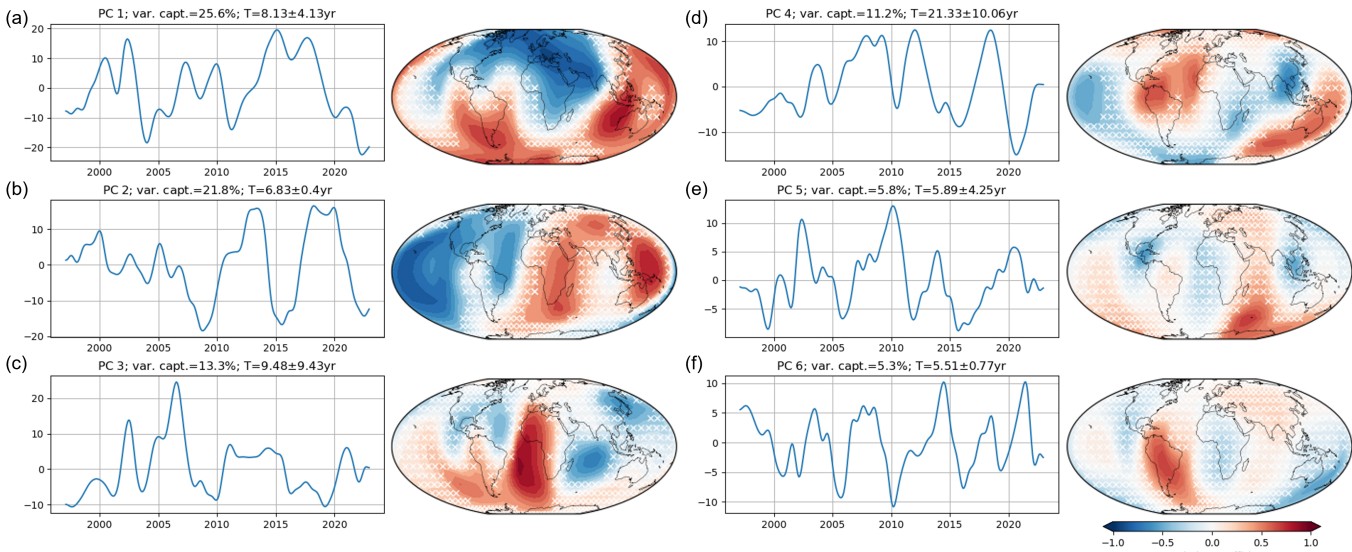

**Figure A3.** PCs and its corresponding spatial correlation pattern of CHAOS-7.12 obtained from PCA





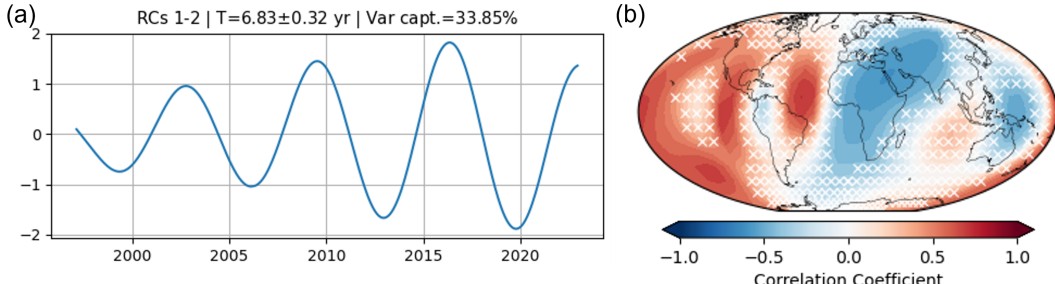

**Figure A4.** (a) Leading S-PC of RC1,2 that creates oscillation of 6.8 year, obtained from MSSA of the CHAOS-7.12. As in Fig. 1, (b) shows the correlation patterns of the mode 6.8 years. The white cross indicates the are with insignificant correlation at the 95% level.





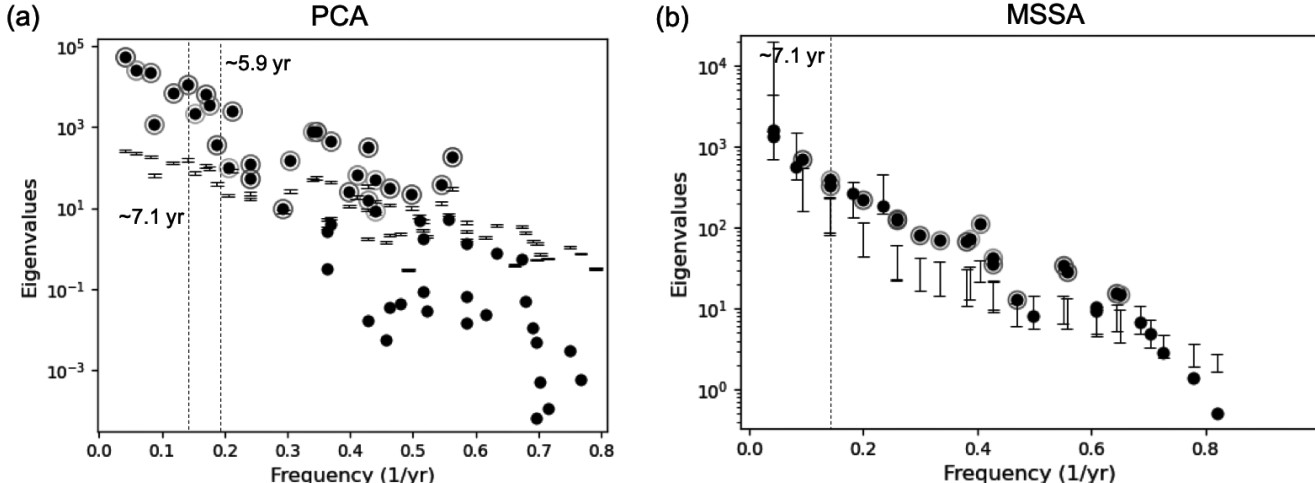

**Figure A5.** Significant test of PCs using Monte Carlo-type hypothesis. (a) Comparison of eigenvalues in PCA analysis between IGG-SLR and surrogates based on AR(p). (b) Spectral properties of IGG-SLR obtained from MSSA, with a subsequent varimax rotation. The estimated eigenvalues are plotted in black dots as a function as their corresponding frequency. The lower and upper ticks on the error bars indicate 5% and 95% of percentiles from a Monte Carlo test with scaled proscrutes target rotation of T-EOFs (Groth and Ghil, 2015). The significant PCs are circled.



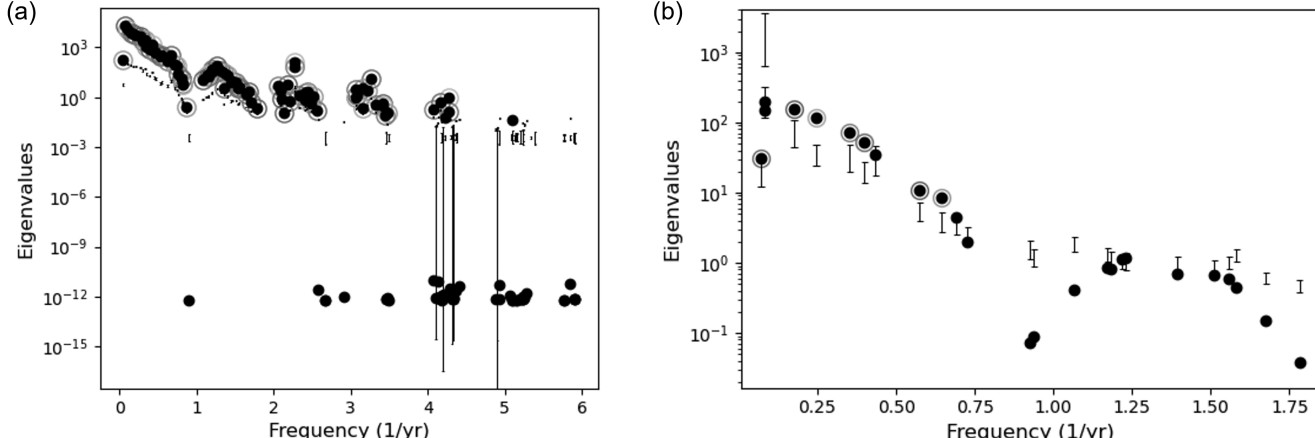

**Figure A6.** Significant test of PCs using Monte Carlo-type hypothesis. (a) Comparison of eigenvalues in PCA analysis between GRACE CSR mascon and surrogates based on AR(p). (b) Spectral properties of GRACE CSR mascon obtained from MSSA, with a subsequent varimax rotation. The estimated eigenvalues are plotted in black dots as a function of their corresponding frequency. The lower and upper ticks on the error bars indicate 5% and 95% of percentiles from a Monte Carlo test with scaled proscrutes target rotation of T-EOFs. The significant PCs are circled.



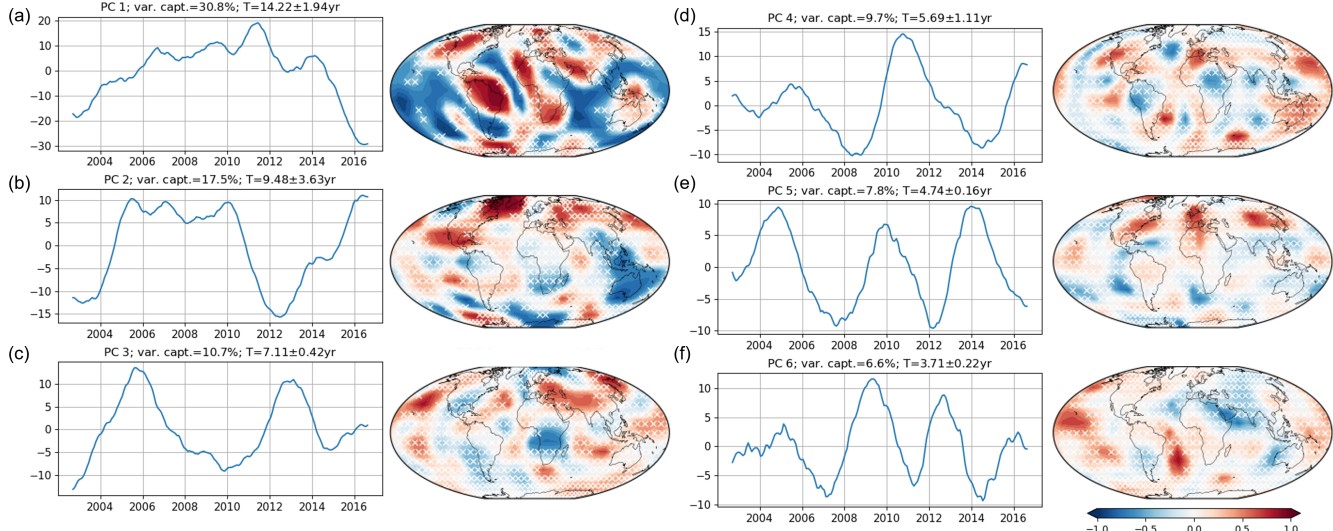

**Figure A7.** PCs and its corresponding spatial correlation pattern of GRACE CSR mascon obtained from PCA





## Appendix B: Joint analysis

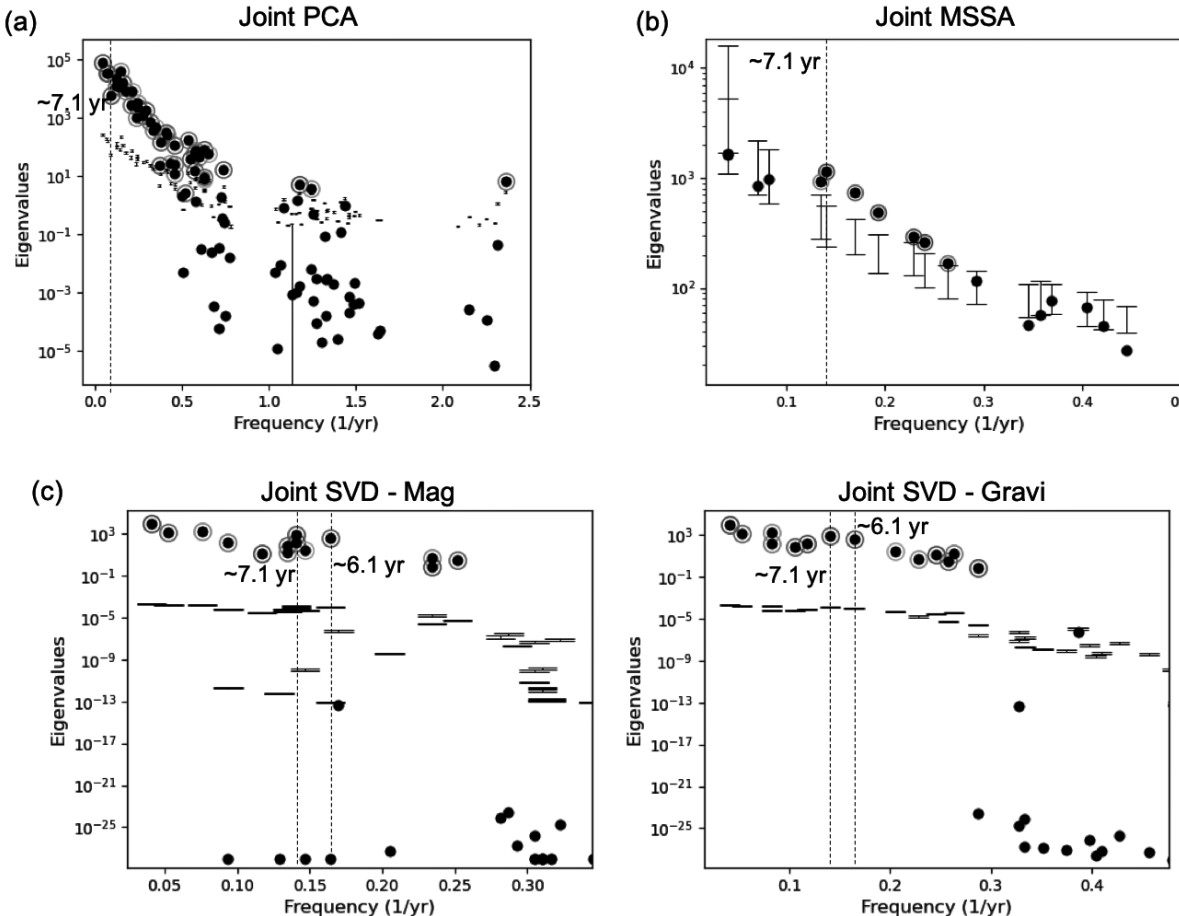

**Figure B1.** Significant test of PCs using Monte Carlo-type hypothesis. (a) Comparison of eigenvalues from coupled PCA analysis of the joint fields and surrogates based on AR(p). (b) Spectral properties of joint fields obtained from MSSA, with a subsequent varimax rotation. (c) Comparison of eigenvalues in SVD analysis between joint fields (COV-OBS.x2 and IGSS-SLR) and surrogates based on AR(p). The estimated eigenvalues are plotted in black dots as a function of their corresponding frequency. The lower and upper ticks on the error bars indicate 5% and 95% of percentiles from a Monte Carlo test with scaled proscrutes target rotation of T-EOFs (Groth and Ghil, 2015). The significant PCs are circled.





**Figure B2.** (a-f) PCs obtained from PCA of the joint field. On the right part, the correlation map of the CHAOS-7.12 and IGG-SLR associated with each PC. The percentage of the variance captured by each PC is shown on the top of the time expansion. The portion of the variance captured in each field is mentioned at the top of the correlation map. The white cross indicates the areas with insignificant correlations at the 95% level.



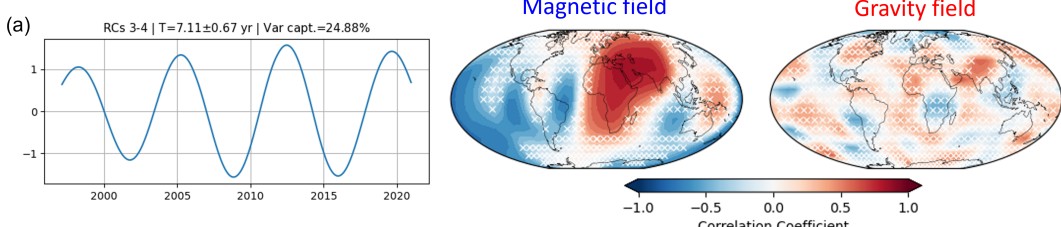

**Figure B3.** Reconstruction of the joint field of oscillatory pairs at a period length of 7.1 year. The correlation patterns of CHAOS-7.12 and IGG-SLR are given on the right side. The MSSA here uses a window length of $M = 110$ months. The white cross indicates the areas with insignificant correlations at the 95% level.





**Figure B4.** (a-f) The first six PCs of the magnetic (blue line) and gravity field (red line) obtained from the joint SVD technique of CHAOS-7.12 and IGG-SLR. The corresponding dominant period is written in the legend. The correlation patterns of the magnetic and gravity field of each mode are given on the right side. The white cross indicates the areas with insignificant correlations at the 95% level.





## Appendix C: Common spatial properties of 6-year mode

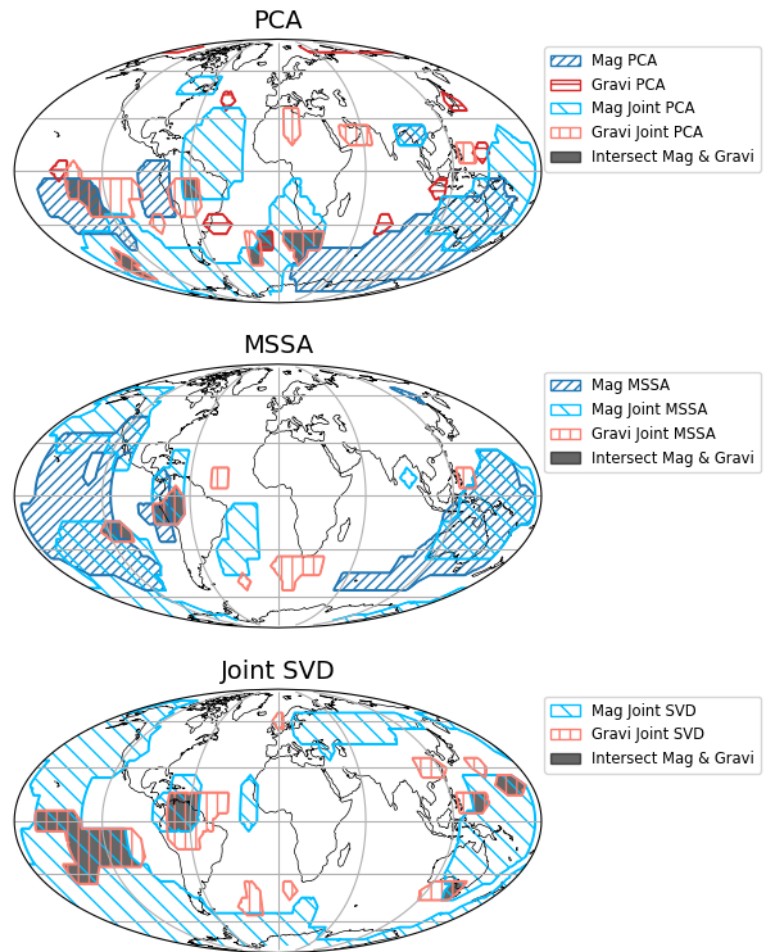

**Figure C1.** Scatters of areas associated with the 6 year mode where the correlation coefficients between the potential fields and the obtained time PC from PCA (a), MSSA (b), and Joint SVD (c) are significant. The layout is the same as for Fig. 9.