# Peer review of "Earth's core variability from the magnetic and gravity field observations"

_EGUsphere, 2023_

## Author Response (AR1)

Dear Editor and reviewers,

Please find our revised manuscript EGUSPHERE-2023-856 entitled "Earth's core variability from the magnetic and gravity field observations".

In this revised version, we paid attention to the remarks of the reviewers, and we have compiled the comments that have been addressed. To ensure readability in our response below, the reviewers' comments are written in black, while our responses are indicated in blue-coloured text.

We hope the revised manuscript meets the quality criteria of the Solid Earth journal.

We thank you and the two anonymous reviewers for the given suggestions and constructive comments.

Best regards,
Anita Thea Saraswati
* * *
**Response to #Reviewer 1**

The study investigates common spatial and time variability of magnetic and gravity fields over interannual to decadal time scales using the most recent models and modern decomposition methods (PCA and derived) and adequate statistical significance tests. The authors highlights some variability modes in both gravity and magnetic field whose periods are in line with core oscillations mentioned in other studies related to core or to Earth rotation. Although the 7-yr mode pattern in the gravity and magnetic field is not particularly revealing in terms of mechanism at this stage, this study shed new lights onto the subtle links between core dynamics and gravity signature via mass transports in the core and encourages to deepen the investigations with new models or even by looking at what happens in the external fluid layers. One strength of this paper is the multiple methods used to validate the robustness of the modes (in particular the joint analysis) and the care brought to derive the statistical significance of all results.

I recommend this paper be published. I have some minor items to suggest.

We thank the referee for the dedicated time and constructive comments that improved our initial manuscript.

1) 2.1 Data: it seems there is no uncertainties associated with the magnetic field and gravity field grids, right? Could the author be clear on this point? As a consequence, the linear trends subtracted to time series in 2.1.1 and 2.1.3 are unweighted least-squares?

We thank you for underlining this aspect. The linear trends are subtracted from the time series using unweighted least-squares. We would like to mention at this stage that we did not take into account the uncertainties of the data. We indicated this aspect in the new version of the manuscript (line 114).

2) It is never clear whether the magnetic field, SV or SA is used.

The SA of the magnetic field is used in this study. We clarify that in the revised manuscript (line 75).

3) In 2.1.1, line 74, 'The first and respectively the second...': although my English is not so good, I think the word 'respectively' is misplaced.

Thank you for the remark. We clarify this sentence in the revised manuscript (line 74) into: "The first and the second derivatives in the radial direction of the core magnetic field are known as secular variation (SV) and secular acceleration (SA), respectively.".

4) 3.1. Although one can refer to fig 8 to see the uncertainties on the periods, it could be interesting to mention the uncertainty on the 7.1-yr and 5.1-yr periods directly in the text (or their order of magnitude).

This has been done. The uncertainty of those periods are mentioned in the section Results.

5) The MSSA uses a window length of 110 months (I found it in the figure caption). How was determined this value?

As stated in Groth, A. & Ghil, M. (2015), the window length (M) is a flexible parameter that is chosen based on the consideration of the analyst and the purpose of the analysis. Ghil et al. (2002) and Weinberg M. D. & Petersen M. S. (2021) also mentioned in their study that the window length should be long enough to cover the anticipated time variation and smaller than half of the data length to yield significant cross-correlation.

Therefore, the window length of 110 months is chosen by considering two aspects. First, the window length of 110 months allows us to capture the expected common interannual signal between the magnetic and the gravity field as long as 9 years. Second, this length is close to one-third of the total length of the time series (338 years), which is still no longer than half of the time series.

Reference:

Ghil, M., Allen, M.R., Dettinger, M.D., Ide, K., Kondrashov, D., Mann, M.E., Robertson, A.W., Saunders, A., Tian, Y., Varadi, F. and Yiou, P., 2002. Advanced spectral methods for climatic time series. Reviews of geophysics, 40(1), pp.3-1.

Groth, A. and Ghil, M., 2015. Monte Carlo singular spectrum analysis (SSA) revisited: Detecting oscillator clusters in multivariate datasets. Journal of Climate, 28(19), pp.7873-7893.

Weinberg, M.D. and Petersen, M.S., 2021. Using multichannel singular spectrum analysis to study galaxy dynamics. Monthly Notices of the Royal Astronomical Society, 501(4), pp.5408-5423.

**Response to #Reviewer 2**

Review of "Earth's core variability from the magnetic and gravity field observations"

This is a good overall analysis of the temporal variations of the magnetic and gravity fields. The results are focused on describing the possible correlation in the temporal and spatial structure of the magnetic and gravity fields. It is overall well written. I do not have major comments or disagreements. I only have a few comments and suggestions that I believe can improve the paper.

We thank the reviewer for taking the time to provide comments that can help improve the manuscript.

— General comments

1)  line 91-93: I agree that that GIA and seismic signals have been extracted from gravity data, but I don't think core processes have, despite the claims of Mandea et al 2012, 2015. These are temporal gravity signatures that have been 'suggested' or 'hypothesized' as due to core processes, certainly not 'evidenced'.

    We changed the mentioned sentence in the revised manuscript to clarify the message (line 93-95): Deeper phenomena, such as core processes and dynamics (Mandea et al., 2012, 2015) are also have been suggested in the temporal gravity signatures.

2)  line 115: What does high-frequency mean here? Why not simply say 'To remove all signals with periods of one year or smaller, …'

    We changed this part in the revised manuscript to "To remove all signals with periods of one year or shorter, …" (line 116).

3)  line 293: '…with consistent spatial patterns, in certain regions at least'. The consistency is only for maps of the same field, e.g. maps of the magnetic field for the 7 yr period do all share a general geographic patterns. But the patterns of the B-field and Gravity field are vastly different for the same mode period. The sentence is a bit ambiguous , as it does not make this distinction.

    We appreciate the reviewer's comment as it accurately reflects the message of the sentence. We clarify this part in the revised manuscript as: "During the 7-year period, the spatial patterns of the magnetic field from all analyses consistently displayed similar general geographic patterns, as do the gravity field maps. Nonetheless, the difference between the magnetic and gravity fields was noticeable, as shown in the analyses of each field." (line 295).

4)  line 337: While I appreciate that 'Building complete models of such motions is beyond the scope of this paper', I feel nevertheless that a little more analysis would be welcomed. For example, the gravity signature of core processes must be transmitted elastically through the mantle before reaching the surface. The associated Love numbers (CMB pressure and density) decrease fast with harmonic degree (e.g. Dumberry & Mandea 2022), so for the same core flow, the core-driven gravity signal is expected to have a larger spatial scale than the magnetic field signal. Yet, the results here show the opposite. I do not think it devalues

the work done here to point out this discrepancy. If indeed the gravity variations highlighted here are caused by core processes, the authors need to point out honestly the challenges associated with this interpretation.

We agree that the expected spatial scale of the gravity signal at the Earth's surface due to the core processes should be larger than what we observed in this study. However, we had mentioned in the previous paragraph, that the spatial patterns in the gravity field obtained in this study might still be influenced by climatic or other dynamics on the Earth's upper layers. This makes it challenging to separate gravity field variations linked to the Earth's core. As we mention in the manuscript, and also mentioned in previous studies, we still have limitations in isolating the signal due to the core dynamics, that hinder us from providing a deeper or complete analysis. However, we appreciate the reviewer's suggestion to provide additional analysis and description of the challenge in the interpretation into the revised manuscript, and propose an additional paragraph (line 336):

"We note that there are still limitations in isolating the signal linked to the core dynamics, that hinder us from providing a deeper or complete analysis. Dumberry and Mandea (2022) pointed out that interpreting the amplitude and spatial pattern of the gravity signal due to the core processes is prone to ambiguity since the resulting signal is relatively weak compared to that from mass anomalies in the crust and mantle. However, detecting core signatures in the gravity field can be relieved through its temporal variations, with a careful analysis of all sources presented in the gravity field. Although the spatial patterns between the two fields show some discrepancies, we consider that the temporal changes in the gravity field align with the signature of core contribution which is observed in the geomagnetic field."

— Typos etc —

1) The title would read better as "Earth's core variability from observations of the magnetic and gravity fields"

2) line 22: mountain torque -> topographic torque

3) line 22: rheology properties inside the core?  I suspect you mean the 'inner core'.

4) line 25: why only the inner core? Why not both the fluid core and inner core?

5) line 34: sensible -> sensitive

6) line 39: to evidence -> to offer evidence of

7) line 60: 'newly' satellite measurements? Do you mean 'recent'?

8) line 120: 'de-normalize them back'??? I have no idea what that means (and that does not sound like proper English to me)

9) line 215: Fig2c -> Fig2b

10) line 234-235: 'from PCA … T=7.1 years' is repeated twice in the same sentence.

We thank the reviewer for the remarks on the typos. We have corrected and improved the above-mentioned comments on the respective line in the revised manuscript.